# Regional activity within the human amygdala varies with season, mood and illuminance

Islay Campbell [1,2,4], Jose Fermin Balda Aizpurua[1,4], Roya Sharifpour[1,3], Ilenia Paparella[1], Elise Beckers [1], Nasrin Mortazavi[1], John Read [1], Christophe Phillips [1], Fabienne Collette [1], Puneet Talwar [1], Laurent Lamalle [1], Mikhail Zubkov[1] & Gilles Vandewalle [1] ✉

The brain mechanisms through which changes in season and light exposure modulate mood may involve different nuclei of the amygdala. We aimed to test this hypothesis using 7 Tesla functional magnetic resonance imaging in 29 healthy young adults. We first considered time-of-year changes in activity that are related to the slow change in photoperiod. We find that the response to emotional stimuli of selected medial and superior nuclei of the amygdala peaked around the start of winter or increased with worse mood status. We further assessed how alternating short exposures to light of different illuminance acutely affected the regional activity of the amygdala. We show that the same areas showed a linear reduction of activity when exposed to increasing light illuminance, specifically when processing emotional stimuli. Importantly, the impact of light on part of these nuclei peaked around the start of summer or decreased with worse mood. These findings provide additional evidence that humans show seasonality and that, for mood, it involves parts of the amygdala. The results bring insights into the mechanisms that underlie the long-term and acute impact of light on mood and that may contribute to the benefits of light therapy in the treatment of mood disorders.

Mood and the symptoms of many psychiatric disorders show seasonal variations[1]. These variations are driven in part by the shorter photoperiod taking place during the fall and winter seasons. Light therapy is established as a non-pharmacological treatment for seasonal affective disorder (SAD), including its subclinical forms and the relatively common winter blues[2,3]. In addition, it is also considered an efficient adjunct therapy for other non-seasonal psychiatric disorders[2–4]. Beyond light's long-term seasonal effects, light can also trigger acute (or immediate) responses that can modulate emotional processing[5]. The daily repetition of these acute

responses is plausibly contributing to the long-term impact of light on mood.

The mechanism of light's influence on mood and emotional processing is most likely mediated through the light-sensitive pathway of the retina that detects environmental irradiance to regulate multiple biological functions not directly related to image formation[6–9], often referred to as the non-image forming (NIF) pathway[10]. Intrinsically photosensitive retinal ganglion cells (ipRGCs) are the main photoreceptors of the NIF system and express the photopigment melanopsin, which is maximally sensitive to blue-wavelength light

[1]GIGA-CRC-Human Imaging, University of Liège, Liège, Belgium. [2]Present address: Sir Jules Thorn Sleep and Circadian Neuroscience Institute, Nuffield Department of Clinical Neurosciences, University of Oxford, Oxford, UK. [3]Present address: Department of Translational Neuroimaging, Hôpital Erasme, Hôpital Universitaire de Bruxelles, Université libre de Bruxelles, Brussels, Belgium. [4]These authors contributed equally: Islay Campbell, Jose Fermin Balda Aizpurua. ✉e-mail: gilles.vandewalle@uliege.be

(~480 nm)[11]. IpRGCs combine their intrinsic photosensitivity with inputs from rods and cones[12] to influence multiple brain targets with a maximal efficiency shifted towards blue-wavelength light[13].

The amygdala coordinates emotional signals[11,14]. It is a complex structure that is divided into several nuclei – likely 13 nuclei in humans, though their exact classification is still debated[15]. Rodent data has shown that the amygdala receives direct inputs from ipRGCs and may play a key role in mediating light exposure behaviour[11]. In rodents, ipRGCs project to the central nucleus of the amygdala and this pathway was found to mediate anxiety-related behaviour after acute light exposure[16,17]. Lesions of the medial amygdala, to which ipRCGs also project, induced altered light-enhanced startle and open-field behaviour, suggesting that the effects of light on anxiety may also be mediated by the ipRGCs innervation of the medial amygdala[18]. In addition, increased blue wavelength light at night was further reported to increase the activity of the basolateral amygdala in rodents[19]. Short photoperiod in rodent models was further reported to modify fear conditioning behaviour while increasing the dendritic spine density of the neurons of the basolateral amygdala[20]. The translation of these acute effects of light in rodents to humans is not fully established.

In vivo neuroimaging studies on the topic in Humans are scarce and have reported contrasting results. Functional Magnetic Resonance Imaging (fMRI) showed that blue-wavelength (473 nm) light enhanced the responses to emotional stimuli in the amygdala as well as its crosstalk with the hypothalamus[21]. In contrast, in a resting-state fMRI paradigm, exposure to a warm polychromatic white light (~2800 K; 100 lux) suppressed amygdala activity compared to darkness (<1 lux) while its connectivity with the ventromedial prefrontal cortex was enhanced[22]. Furthermore, a dose-dependent effect of three weeks of bright-light therapy was reported to reduce threat-related reactivity of the amygdala and to increase its functional connectivity with medial prefrontal cortex, suggesting these changes may be part of the mechanism that mediates the beneficial effects of bright-light therapy[23]. The difference between studies may arise from differences in the light sources, from being engaged in a cognitive task vs. at rest and from considering shorter or longer impacts of light. They may also be due to the context-dependent response of the different nuclei of the amygdala that could not be addressed, given the data resolution. Moving away from light's impact, seasonal differences in the volume of subregions of the human amygdala were reported, with a peak in the summer. However, there was no association between mood measures and amygdala subregions volumes and photoperiod[24].

Overall, animal models and human research suggest the amygdala is a candidate to mediate part of the impact of light exposure, photoperiod, and season have on mood. Given the known involvement of the amygdala nuclei in psychiatric disorders[25–27], a precise understanding of the mechanisms at play may have implications beyond the healthy individuals and could help to improve the management of seasonal symptoms in psychiatric disorders, the use of light therapy and/or extend it to more brain disorders.

Here, we aimed to determine whether the regional activity of the amygdala varied with time-of-year and whether light exposure affected the activity of its different nuclei. We took advantage of the higher resolution of ultra-high-field (UHF) 7 Tesla (7T) fMRI to record the activity of the amygdala in healthy young adults (devoid of psychiatric disorders) exposed to the light of various illuminances while engaged in an auditory emotional task that was previously successfully used to show an influence of light on the activity of the amygdala[21]. We find that the regional activity of four subparts of the medial and superior amygdala shows an association with time-of-year or mood, and with the acute impact of illuminance, specifically when processing emotional stimuli. In addition, the impact of light on part of these nuclei varied with time-of-year and mood. These findings provide insights into the mechanisms that may underlie the impact of photoperiod and light exposure on mood and psychiatric symptoms.

## Results

The brain activity of 29 healthy young participants (24 y ± 3.1; 18 women) was recorded during an fMRI protocol conducted in Liège (Belgium) in the morning (scans 3–3.5 h after habitual wake-up time), and including three different cognitive tasks, one of which is an emotional auditory task and the sole focus of this analysis (Fig. 1, Supplementary Table S1). During the task, participants were alternatively maintained in darkness or exposed to different light consisting of a blue-enriched polychromatic white light of three different illuminance levels (37, 92, 190 melanopic Equivalent Daylight Illuminance – mel EDI - lux) and a low illuminance monochromatic orange light (0.16 mel EDI lux).

Participants achieved 92 ± 7.3% (mean ± SD) button response, while the accuracy to the lure gender classification task[28] was 79.7% ±9.9%, which is slightly lower than previous studies using the same task[29]. Critically, the reaction times were faster for the neutral (1203 ± 183 ms) as compared with the emotional stimuli (1249 ± 184 ms) (main effect, stimulus type; $F_{1,28} = 14.6$, $p = 0.0007$) (Fig. 2C). This is in line with previous literature[5,29] and confirms that the emotional content of the stimuli was successful in triggering a behavioural response. Reaction times were not significantly affected by illuminance (main effect, illuminance; $F_{4,221} = 0.92$, $p = 0.45$) or by time-of-year (using the cosine value associated with day of the year over a year-long cycle, see methods; main effect, time-of-year; $F_{1,25} = 0.12$, $p = 0.73$) and there was no interaction between illuminance and stimulus type ($F_{4,221} = 0.35$, $p = 0.85$). This does not, in principle, preclude illuminance or time-of-year from affecting the underlying brain activity and ensures that neuroimaging findings related to light exposure are not merely the results of a change in reaction times. We further assessed variation in the mood of our participants, based on scores on the Beck Depression Inventory-II[30], and found no association with time-of-year, age, sex and BMI ($F_{1,20} < 0.65$, $p > 0.4$). The questionnaire-based seasonality score[31] was low in all participants and was not used in the analyses (Supplementary Table S1).

### The activity of the medial-superior amygdala region is related to time-of-year and mood

We used a template that consistently divided the amygdala into 10 subparts[32] (Fig. 2A) to extract the regional activity within the amygdala. The statistical analysis, which controlled for age, sex and BMI, first confirmed that the task was successful in triggering emotional responses in the amygdala with higher response to emotional than to neutral stimuli (GLMM; main effect, stimulus type, $F_{1, 27.74} = 9.93$; $p = 0.0039$, $R^{2*} = 0.26$; Fig. 2D) (Supplementary Table S3). This is supported by the visualization of the whole brain group analysis ($p_{uncorrected} < 0.001$) which yielded local positive peaks bilaterally in the amygdala (mostly over the medial and superior subparts – -subparts 1, 3, 5 and 10) indicating that our finding does not arise from a nearby "leaking" activation (Fig. 2B). As suggested by this visualization, responses significantly differed across amygdala subparts (GLMM; main effect, subpart, $F_{9, 499.9} = 2.13$; $p = 0.0256$, $R^{2*} = 0.04$; Fig. 2E), though not in interaction with stimulus type (GLMM; interaction, stimulus type by subpart: $F_{9, 499.9} = 0.56$, $p = 0.85$) (Supplementary Table S3). Post hoc tests indicated that three subparts did not respond significantly to the auditory stimulations ($p > 0.05$, Supplementary Table S3; Intermediate and dorsal basolateral nucleus - subpart 2, blue on Fig. 2A, Central nucleus - subpart 4, purple on Fig. 2A, and Ventral basolateral nucleus and Paralaminar nucleus - subpart 6, red on Fig. 2A). The remaining of the analyses only considered the 7 other subparts.

Although low mood was among our exclusion criteria, it varied across our participants, which were recorded at all times of the year (Winter: 4; Spring: 10; Summer: 6; Fall: 9; Supplementary Table S1). We therefore asked whether the activity of the amygdala subparts was associated with the time-of-year and mood. The statistical analysis

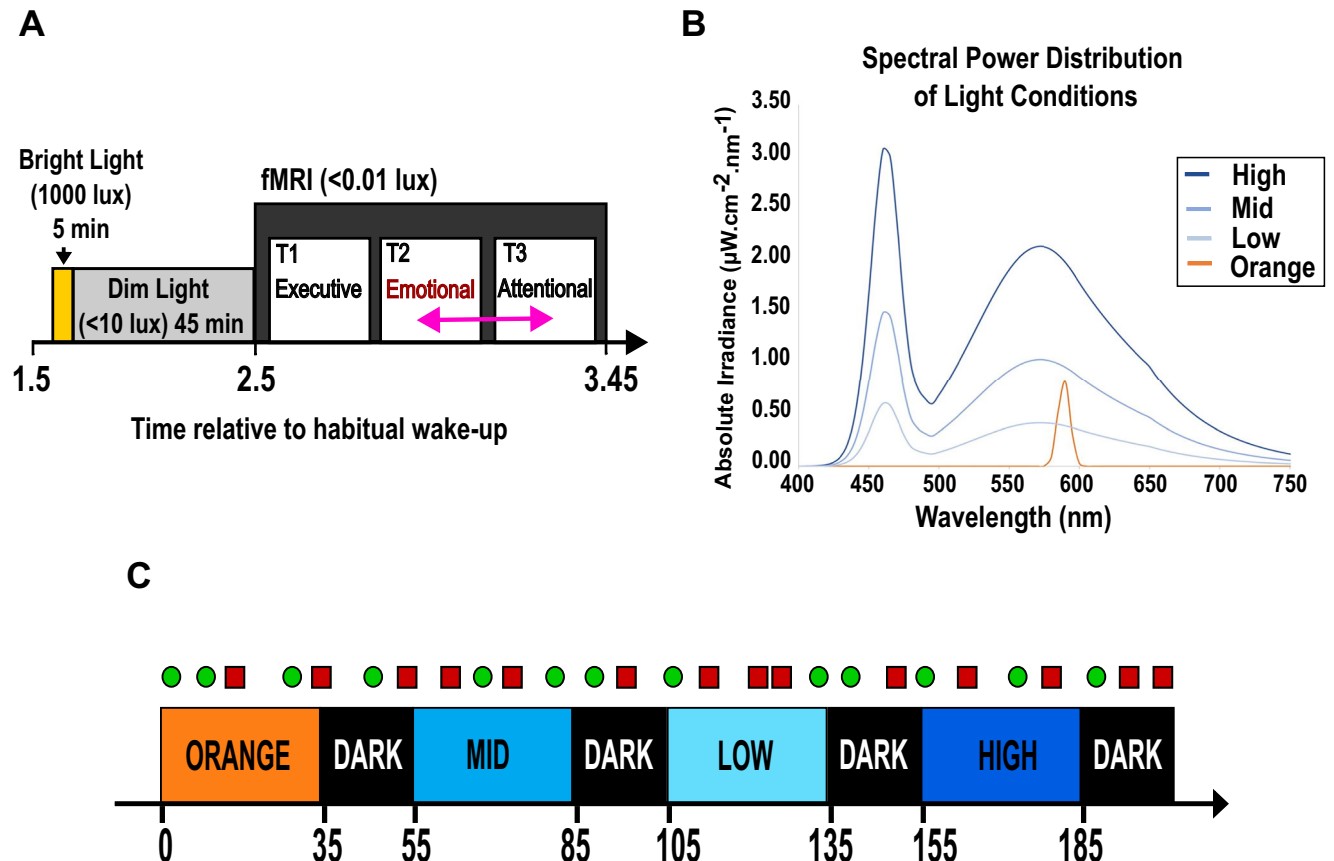

**Fig. 1 | Experimental protocol. A** Overall timeline. Participants followed 7-days of regular loose sleep-wake schedule (+− 1 h, verified using actigraphy) before the MRI scan. Participants arrived at the laboratory 1.5 h after wake-up time. The prior light history of participants was standardised on the morning of the MRI scan, consisting of 5-min 1000 lux light and the following 45-min under dim (<10 lux) light. In total, 29 participants (24 y ± 3.1; 18 women) performed an executive (always first), an emotional and an attentional task (pseudo-randomly 2nd or 3rd, pink arrow; meaning that the fMRI recording of the emotional task was completed 3–3.5 h after wake-up time). Only the emotional task is considered in the present manuscript. The protocol was administered at different times of the year and completed once by each participant (Winter: 4; Spring: 10; Summer: 6; Fall: 9; Supplementary Table S1). **B** Spectral power distribution of light exposures. Monochromatic orange: 0.16 mel EDI lux; Polychromatic, blue-enriched light (6500 K); LOW, MID, HIGH: 37, 92, 190 mel EDI lux (Supplementary Table S2 for full details). Blue-enriched illuminances were set according to the technical characteristics of the light source and to keep the overall photon flux similar to prior 3 T MRI studies of our team[5,74]. The orange light was introduced as a control visual stimulation for potential secondary whole-brain analyses. For the present region of interest analyses, we discarded colour differences between the light conditions and only considered illuminance as indexed by mel EDI lux, constituting a limitation of our study. **C** Emotional task and light blocks. The task consisted of a lure gender classification of meaningless vocalisation[29]. Untold to the participants, vocalisations were pronounced with neutral (50%, green circle) or angry/emotional (50%, red square) prosody by professional actors (50% women). Participants were pseudo-randomly exposed to the four light conditions or a control darkness period. Time is reported in seconds relative to session onset.

provided a positive answer to the question (GLMM; main effect, time of year: $F_{1, 17.85} = 12.6$, $p = 0.0023$, $R^{2*} = 0.41$; main effect, mood: $F_{1, 17.8} = 17.5$, $p = 0.0006$, $R^{2*} = 0.495$) and further revealed that the association with time-of-year and mood differed across amygdala subparts (GLMM; three-way interaction, subpart by time of year by mood: $F_{6,272.4} = 4.03$, $p = 0.0007$, $R^{2*} = 0.08$; interaction, subpart by time of year: $F_{6, 272.6} = 4.66$, $p = 0.0002$, $R^{2*} = 0.09$; interaction, subpart by mood: $F_{6, 272.4} = 2.99$, $p = 0.0076$, $R^{2*} = 0.062$) (Supplementary Table S4.a).

Considering first time-of-year, post hoc tests showed that the activity of the basomedial nucleus (BMN; $t = 3.54$, $p_{corrected} = 0.004$, subpart 3, dark blue on Fig. 2A) and of the anterior amygdaloid area (AAA; $t = 4.96$, $p_{corrected} < 0.0001$, subpart 9, light green on Fig. 2A) significantly varied with time of year (both subparts were also different from part of the other subparts in their association with time-of-year) (Fig. 2F, G, Supplementary Table S4.b,c). For both areas, responses were stronger around the start of winter (Fig.2H, I). Post hoc tests then revealed that worse mood was significantly associated with a higher activity of medial and cortical nuclei (MCN; $t = 4.28$, $p_{corrected} = 0.0002$, subpart 5, violet on Fig. 2A) and amygdala transition areas (ATA,

$t = 4.64$, $p_{corrected} < 0.0001$, subpart 7, orange on Fig. 2A) (both subparts were also different from part of the other subparts in their association with time of year) (Fig. 2J, K, Supplementary Table S4.d, e). A statistical trend for an association with mood was further detected for the AAA ($t = 2.6$, $p_{uncorrected} = 0.011$, $p_{corrected} = 0.061$, subpart 9, light green on Fig. 2A) and the intercalated nuclei ($t = 2.46$, $p_{uncorrected} = 0.016$, $p_{corrected} = 0.089$, subpart 10, green on Fig. 2A) (Fig. 2L, M, Supplementary Table S4.d). Overall, the results suggest the activity in subparts 3 and 9 varied significantly with the time-of-year, while activity in subparts 5 and 7 varied significantly with mood. Thus, activity in selected nuclei comprising the amygdala's medial and superior region is either related to time-of-year or mood.

**Light illuminance linearly decreases the activity of the medial-superior amygdala**

The next statistical analysis included the impact of illuminance variation on the activity of the amygdala subparts, still controlling for age, sex and BMI. It revealed that the illuminance affected differently the activity of the different subparts (GLMM; main effect, subpart: $F_{6, 331.4} = 7.05$, $p < 0.0001$, $R^{2*} = 0.11$) (Fig. 3B, Supplementary Table S5).

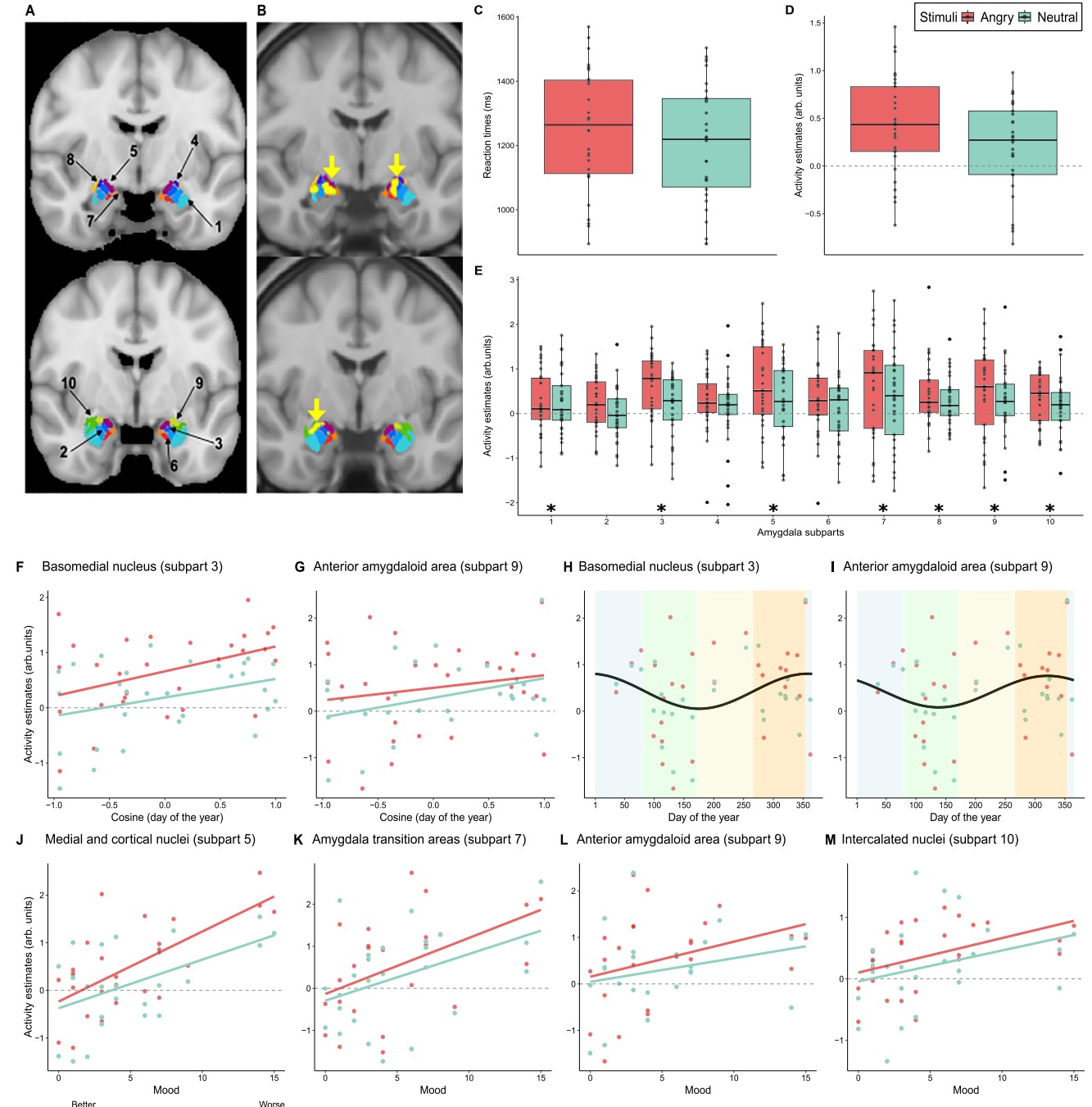

The impact of light further depended on the stimulus types (GLMM; main effect, stimulus type: $F_{1, 27.2} = 6.32$, $p = 0.0181$, $R^{2*} = 0.19$), while there was no interaction between stimulus type and subparts (GLMM; interaction, subpart by stimulus type: $F_{6, 331.4} = 1.42$, $p = 0.21$) (Supplementary Table S5).

Post hoc analyses found that the BMN ($p_{corrected} = 0.0001$, subpart 3, dark blue, Fig. 2A), medial and cortical nuclei ($p_{corrected} < 0.0001$, subpart 5, violet, Fig. 2A), ATA ($p_{corrected} = 0.0005$, subpart 7, orange, Fig. 2A) and the AAA ($p_{corrected} = 0.0002$, subpart 9, light green on Fig. 2A) were significantly affected by illuminance changes, and differently so compared to many other amygdala subparts (Supplementary Table S5). Visualization of the whole brain group analysis ($p_{uncorrected} < 0.001$) yielded local negative peaks bilaterally in the amygdala when comparing emotional and neutral items, mostly over the left subparts 3, 5 and 7 and the right subparts 3, 5 and 9, again supporting the finding and indicating that it does not arise from a nearby "leaking" activation (Fig. 3A). When investigating in more detail the regional impact of light in the 4 subparts, results are compatible with a linear decrease in activity with increasing illuminance in the 4 subparts (Fig. 3C, D; Supplementary Table S6). The regions that were related to time-of-year and/or mood independently of illuminance are therefore also linearly affected by illuminance variations when processing emotional stimulations.

## The regional impact of light on the amygdala depends on time-of-year and mood

In the final steps, we asked whether the impact of illuminance on the different amygdala subparts varied with the time of the year and mood. The statistical analysis (controlling for age, sex and BMI), showed that the association with time-of-year and mood differed across amygdala subparts (GLMM; interaction, subpart by time-of-year: $F_{6, 272} = 2.94$, $p = 0.0085$, $R^{2*} = 0.06$; interaction, subpart by mood: $F_{6, 272} = 2.24$,

**Fig. 2 | Regional difference in the response of amygdala subparts to emotional and neutral auditory stimuli and relationships with time-of-year and mood.** For all panels displaying data points the sample size is 29. Source data are provided as a Source Data file. **A** Segmentation of the amygdala into ten subparts. The amygdala template isolate nuclei and nucleus groups as follows: (1) Lateral nucleus, (2) Intermediate and dorsal basolateral nucleus, (3) Basomedial nuclei, (4) Central nucleus, (5) Medial and Cortical nuclei, (6) Ventral basolateral nucleus and Para-laminar nucleus, (7) amygdala transition area composed of Amygdalocortical area, Amygdalohippocampal area, Periamygdaloid cortex, (8) Amygdalostriatal, (9) Anterior amygdaloid area, (10) Intercalated nuclei[32]. **B** Visualisation of the whole brain analyses of the difference between emotional and neutral auditory stimuli over the amygdala area. A local positive peak (yellow arrow and yellow cluster; $p_{uncorrected} < 0.001$) was detected bilaterally in the amygdala mostly -over the subparts 1, 3, 5 and 10. These results indicate that our finding does not arise from a nearby "leaking" activation. **C** Significant difference in reaction times to neutral and angry stimuli. Reaction times were faster for the neutral compared with the emotional stimuli (main effect, stimulus type; $F_{1,28} = 14.6$, $p = 0.0007$). **D, E** Impact of stimuli type on the activity of the amygdala subparts. Activity estimates (arbitrary unit – arb. units) averaged over the entire amygdala (**D**) and for each subpart (**E**). **D** The amygdala has a higher response to emotional than to neutral stimuli (GLMM;

main effect, stimulus type, $F_{1, 27.74} = 9.93$; $p = 0.0039$, $R^{2*} = 0.26$). **E** Amygdala subparts respond significantly different (GLMM; main effect, subpart, F9, 499.9 = 2.13; $p = 0.0256$, $R^{2*} = 0.04$). Refer to panel A for names corresponding to subpart numbers and refer to Supplementary Tables S3 for full statistics. Black stars indicate amygdala subparts with significant differences. **F–I** Time-of-year variation in the activity of the amygdala subparts. Activity estimates in the basomedial nucleus (**F**; subpart 3) and anterior amygdaloid area (**G**; subpart 9) vs. cosine of degree transformation of day of year ($0° = $ January 1st; 1 day $= 0.98°$; i.e. the value included in the GLMMs) and vs. day-of-the-year for visualisation - basomedial nucleus (**H**; subpart 3) and anterior amygdaloid area (**I**; subpart 9). Refer to Supplementary Table S4 for full statistics. Cosine fits are included for display purposes only and do not replace the outcomes of the GLMMs. Seasons represented by background colours (winter – blue, spring – green, summer – yellow, autumn – orange) for (**H**) and (**I**). **J–M** Association between mood and the activity of the amygdala subparts. Activity estimates in the medial and cortical nuclei (**J**; subpart 5), and in the amygdala transition areas (**K**, subpart 7) was higher when mood was worse according a questionnaire score (Beck Depression Inventory[30]). A similar statistical trend (that did not survive correction for multiple comparisons) was detected in the anterior amygdaloid area (**L**; subpart 9) and the intercalated nuclei (**M**; subpart 10). Refer to Supplementary Table S4 for full statistics.

---

$p = 0.04$, $R^{2*} = 0.05$; three-way interaction, subpart by time-of-year by mood: $F_{6,272} = 0.87$, $p = 0.52$) (Supplementary Table S7). Post hoc tests did not point to a subpart significantly driving the associations but rather yielded statistical trends suggesting that the impact of illuminance varied with time of year within the ATA ($t = -1.73$, $p_{uncorrected} = 0.096$, $p_{corrected} = 0.32$, the subpart differed significantly from several other subparts) while it depended on mood in the AAA ($t = 2.21$, $p_{uncorrected} = 0.033$, $p_{corrected} = 0.12$, the subpart differed significantly from several other subparts) (Supplementary Table S7). Visualisation of the data indicates that light impact on the ATA is maximum around summer (Fig.3E, F), while the worse mood is associated with a larger impact of illuminance on the AAA (Fig.3G).

## Discussion

Cognitive brain function varies with seasons, and psychiatric symptoms in patients also show seasonal variations[1,33]. Light therapy is further established as an effective treatment for mood disorders[2,3]. However, the brain mechanisms underlying seasonality and light impacts have not been fully resolved, though the nuclei of the amygdala are arguably involved[34]. Considering the connections between the neural pathways involved in emotional and mood regulation[35], we aimed to determine which subparts of the amygdala were affected by time-of-year and light during an emotional task using fMRI and an automatic parcellation of the amygdala in 10 subparts (corresponding to a unique nucleus or several nuclei).

Our analysis was twofold: we first considered the seasonal changes (time-of-year) in activity, which are related to the slow change in photoperiod, before assessing how alternating short exposures to light of different illuminance acutely affected the regional activity of the amygdala. We found that the activity in selected medial and superior nuclei of the amygdala was related to either time-of-year or mood. We further found that the same areas showed a linear acute reduction of activity with increasing illuminance, specifically when processing emotional stimuli. Importantly, the impact of light on part of these nuclei varied with time-of-year or with mood. These findings provide additional evidence of seasonality in humans[33,36]. They further bring insights into the mechanisms that underlie the long-term and acute biological impact of light on the brain and that may contribute to the benefits of light therapy in the treatment of mood disorders.

The task we administered was successful in triggering a differential response between neutral and emotional stimuli in most of the subparts of the amygdala isolated by our segmentation procedure. Among these, the response of the BMN and of the AAA is higher around the winter solstice as compared with the summer solstice, at

least in the morning and in the context of our experiment. It is therefore tempting to speculate that the time-of-year variations of these subparts may underlie part of the seasonal variations in mood in healthy individuals or in patients. In addition, although mood did not show significant seasonal variation in our sample, responses were higher when mood was worse within a subpart of our template encompassing the medial and cortical nuclei (as their boundaries are not well-defined enough on MRI images to separate them)[32] and the ATA, which includes several small nuclei.

The BMN and medial nuclei send reciprocal projections to each other, and both project to the cortical nucleus[37]. According to a simplified overview of sensory information flow through the amygdala, information enters through the BMN and then progresses through to the cortical and medial nuclei, which act as an output stations for downstream targets[14]. Activation of the medial nucleus is associated with stress and leads to the secretion of adrenocorticotropic hormone and the activation of the hypothalamic-pituitary axis (a neuroendocrine system that maintains physiological homeostasis)[14]. The medial nuclei mainly contain GABAergic neurons[38] suggesting that it fulfils its functions mainly through an inhibitory influence on downstream areas, within and outside the amygdala.

The AAA, which is less developed in primates, could be among the downstream targets of the medial nucleus, although it receives projections from the lateral and central nuclei[14,37]. Likewise, the amygdalohippocampal area and the periamygdaloid cortex, which compose part of the ATA (together with the Amygdalocortical area), both receive projections from the BMN and medial nuclei and projects back to the medial nuclei[37], such that the ATA, which has received little scientific focus, could be among the downstream targets of the BMN. We speculate, therefore, that the BMN is in an ideal position to affect multiple amygdala-dependent processes, including the seasonal regulation of mood, through the medial and cortical nuclei and ATA. This speculation is only partially supported by our data as activity of the AAA, but not of the BMN, was weakly associated with mood in our data (the association was significant prior to correction for multiple comparisons). This potential scenario may be due to the healthy nature of our sample of a relatively limited size, warranting further investigation in larger and more diverse samples.

Interestingly, the same four subparts that are related to either time-of-year or mood were also acutely affected by ambient light illuminance. Research in rodents reported that the central and medial amygdala receive direct functional inputs from ipRGCs, and both projections were implicated in the impact of light on anxiety-related behaviours[16–18,39,40]. Neuroimaging research in humans reported that

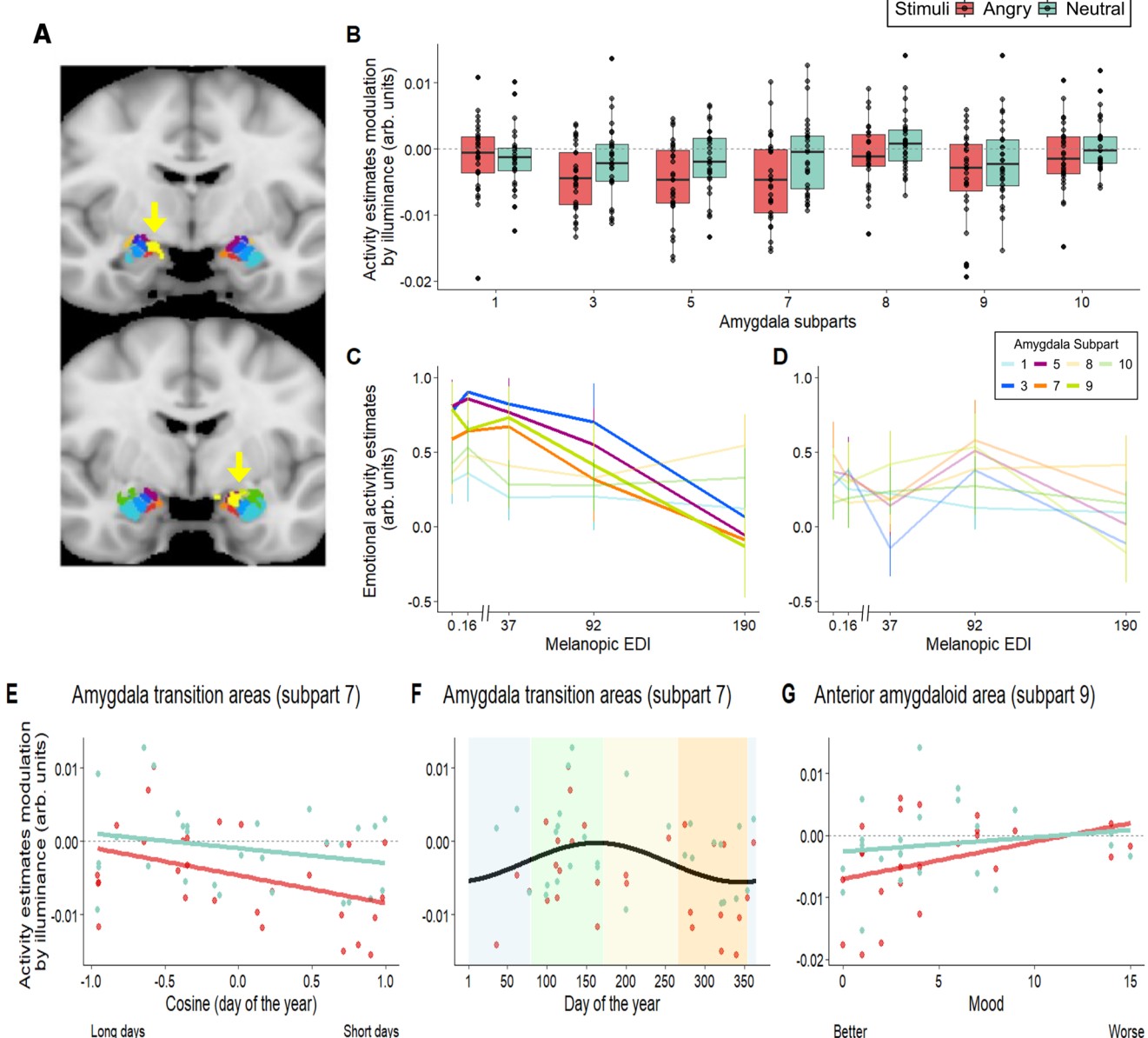

**Fig. 3 | Regional difference in the impact of light illuminance on the responses of amygdala subparts and relationships with time-of-year and mood.** For all panels displaying data points the sample size is 29. Source data are provided as a Source Data file. **A** Visualisation of the impact of illuminance change over the whole brain of the entire sample. Visualisation of the whole brain analyses of the collective impact of the emotional variations in illuminance over the amygdala area. A local negative peak (yellow arrow and yellow cluster; $p_{uncorrected} < 0.001$) was detected bilaterally in the amygdala, mostly over the subparts 3, 5 and 7 in the left hemisphere and over subparts 3, 5 and 9 in the right hemisphere. These results indicate that our finding does not arise from a nearby "leaking" deactivation. **B** Collective impact of illuminance variation on the activity of the amygdala subpart. Parametric modulation of the activity estimates (arbitrary unit - arb. units) by the variations in illuminance. Refer to panel Fig. 2A for names corresponding to subpart numbers and refer to Supplementary Table S5 for full statistics. Illuminance affected the activity of the amygdala subparts differently (GLMM; main effect, subpart: $F6$, $331.4 = 7.05$, $p < 0.0001$, $R^{2*} = 0.11$). **C, D** Impact of each illuminance on the activity of the amygdala subparts. Results are displayed for emotional (**C**) and neutral (**D**) stimuli for all 7 subparts that responded to the task (cf. Fig. 2E). The small inset

provides the number corresponding to the colour code. The four subparts showing a significant impact of the collective change in illuminance are highlighted on panel **C** (only those 4 regions were used for statistics; the other three are shown for comparisons). Refer to panel Fig. 2A for names corresponding to subpart numbers and refer to Supplementary Table S6 for full statistics. **E, F** Time-of-year variation in the impact of light on the activity of the amygdala transition areas (subpart 7). Activity estimates of the ATA vs. cosine of degree transformation of day of year (0° = January 1st; 1 day = 0.98°) (i.e. the value used in the GLMMs) (**E**) and vs. day of the year for visualisation (**F**), as indicated by a statistical trend (not surviving correction for multiple comparisons). Refer to Supplementary Table S7 for full statistics. Seasons represented by background colours (winter – blue, spring – green, summer – yellow, autumn – orange) for (**F**). **G** Association between mood and the impact of light on the anterior amygdaloid area. Activity estimates were higher in the AAA when mood was worse, according to the questionnaire score (Beck Depression Inventory[30]) as indicated by a statistical trend (that did not survive correction for multiple comparisons). Refer to Supplementary Table S7 for full statistics.

light can influence the activity of the amygdala, including during emotional processing, but did not isolate the nuclei involved[5,22]. Our results could suggest that in humans, the medial nuclei mediates the impact of light on emotional state. This provides support for the existence of a functional projection from ipRGCs to the medial parts of the amygdala in humans, similar to rodents[11]. This would mean that, like mice, light can influence emotional status independent of the suprachiasmatic nucleus (SCN), which receives the strongest inputs from ipRGCs based on studies in animals and is the site of the principal circadian clock[11,41].

Our data is also compatible with the effect of light on the BMN and medial nucleus activity through the SCN or other nuclei. According to data collected in rodents, several nuclei of the amygdala (including the 4 subparts we isolated) receive inputs from the orexinergic neurons of the lateral hypothalamus[42] which is known to promote wakefulness as part of the ascending activating system[43]. In rodents, it receives direct projection from ipRGCs[39] and may be affected by illuminance in humans[44]. Light may impact mood through both the direct and indirect routes. Testing whether the timescale (acute vs long-term photoperiod) changes the pathways involved could be important for determining how light affects mood. The acute impact could be mainly mediated by a direct projection to the medial amygdala. In contrast, since seasonal affective disorder was found to be associated with a misalignment of the circadian clock that can be corrected by light therapy[1,45], the longer-term impact of photoperiod could indirectly reach the BMN through the SCN and/or the lateral hypothalamus. Future studies, including connectivity analyses, should test these hypotheses.

Importantly, the impact of light on the amygdala was not uniform across its subparts at the different times of the year and with respect to mood (as indicated by significant interaction terms), at least when light is delivered in the morning. Statistical trends potentially pointed towards the nuclei driving these non-uniform responses to light. The variation with time-of-year may be driven by the ATA, which showed a larger impact of light around the summer solstice, while the variation related to worse mood may arise from the AAA. Future research should test whether these non-uniform responses to light contribute to the effectiveness of light therapy. In addition, the results indicate that the short procedure to standardise recent light history included in our protocol prior to the MRI did not eliminate all time-of-year, implying that a "memory" for longer light history or a true endogenous seasonal/yearly oscillator may underlie our findings[46].

Given the decreased activity and GABAergic activity of the medial nucleus mentioned above[38], one could postulate that the decrease in activity induced by light over the medial and superior nuclei of the amygdala triggers a reduction or inhibition of this inhibitory influence. The hypothesis that ipRGCs have the potential to be inhibitory is not without evidence, as a rodent study found that a subset of ipRGCs release the inhibitory neurotransmitter GABA at brain targets (e.g. SCN), leading to reduced sensitivity of the pupil to light and circadian photoentrainment[47]. Overall, although we do not have empirical data to support this hypothesis, we tentatively speculate that, given the high interconnectivity of the amygdala nuclei, the deactivation in the ATA and AAA in response to illuminance change may reflect the downstream propagation of signals from the output of the BMN and medial/cortical nuclei.

Ultimately, although we favour an impact of light on the amygdala through a direct or indirect projection of the retina to the medial nuclei, all four subparts could equally contribute to the decrease BOLD signal we detect under higher illuminance[37]. Given the changes in functional connectivity previously reported following light therapy when considering the amygdala as a whole[23], part of our results may be mediated by an impact of light on the prefrontal cortex or on cross-talk with the prefrontal cortex. In addition, our findings support that the reduced responsiveness of (part of) the amygdala does not require a

week of light therapy and is already present acutely, during the exposure. The sizes of the effects we detected are small, such that light exposure may be beneficial to the emotional state through a repeated effect on the BMN and medial parts of the amygdala (or on the prefrontal cortex). Again, complex connectivity studies considering light exposure at different timescales are required to test these hypotheses and to assess how our findings fit within a larger network of (most often) small brain regions[48,49].

The decrease of activity we find in several subparts of the amygdala contrasts with previous research done using 3T fMRI in healthy participants, which found that exposure to light increased the response of the amygdala to emotional stimuli using the same emotional task[5]. The conflicting findings may be due to the differences in light sources used as it was a monochromatic blue light (corresponding to an illuminance of ~22 and 93 mel EDI lux). In contrast, another 3T resting state MRI, which did not involve a cognitive task, found that the activity of the amygdala was decreased under polychromatic white light (~35 mel EDI lux; 2800k) in comparison to darkness[22]. In the present study, we used a polychromatic white, blue-enriched light source of three different intensities (37, 92, 190 mel EDI; 6500k), i.e. the lowest light level we used has a similar mel EDI lux to the resting-state study. It may be that the amygdala sees its activity increase or decrease in response to light depending on illuminance and spectrum, depending potentially on the type of retinal photoreceptor recruited.

The BMN, medial nuclei, AAA and the periamygdaloid cortex project downstream to the frontal cortex[37], which governs cognition. In addition, one of the main targets of the medial nucleus is the hypothalamus, specifically over the anterior paraventricular nucleus. The cortical nuclei also project to the brainstem locus coeruleus, producing norepinephrine and the dopaminergic ventral tegmental area[37,42]. Light could affect behaviour both acutely and in the long run through some or all of these projections. We stress, however, that although the task we used[5,28,29] was successful in triggering acute behavioural responses to the emotional content vocalisations when considering reaction times, these were not significantly different across the light exposures of different illuminance. How our fMRI findings explain how longer/repeated light exposures may acutely affect behaviour is not straightforward, and we suspect that longer light exposures and/or other cognitive tasks would be required to trigger clear acute behavioural responses[10].

In terms of photoreceptors, we cannot conclusively say that the ipRGCs are underlying the decrease in activity in the subparts we see at higher light levels. Given the projections of ipRGCs to the medial amygdala and to the hypothalamus reported in rodents[39] and the light level we used, their implication is very likely. Rods and cones are, however, also likely contributing – e.g. as a normalisation of their function was suggested following light therapy[45,], and there may be differential recruitment of photoreceptors at different illuminance levels[50]. Future studies in humans could determine the photoreceptor underlying the impact of light on emotion processing and mood by using metameric light exposures to selectively affect one photoreceptor type[51]. Our protocol bears other limitations, including the inherent non-ecological experiment condition of an MR apparatus, the lack of investigation at other times of day and other light spectra when it is established that time of day and light content in short wavelength photons affect the acute and longer-term biological NIF responses to light[10]. While compatible with usual indoor levels, the light we administered was far from outdoor levels, and we can only speculate that the effects we report are similar under these outdoor conditions. The effects of outdoor light conditions may be stronger, but the light adaptation processes of the retina, in part driven by ipRGCs, may mitigate differences in absolute illuminance[52]. Our protocol also only included negative (and neutral) valence vocalisation which fits well with the important involvement of the amygdala in negative affect[14,15]. Positive stimulation should be included in the future to make

investigating the relationship with the positive effect of light therapy on depressive symptoms more straightforward. Furthermore, we used Beck Depression Inventory-II questionnaire to measure mood in a young healthy population, implying non-clinical variations in mood scores. We cannot generalise our results to populations with depression or other psychiatric conditions.

Emotional regulation is vital and evolutionary critical. The amygdala fulfils part of this regulation to allow for adaptation to the changing environment. We find, in a healthy sample, that several subregions of the amygdala showed seasonal variation in activity as well as reduction of activity with increasing illuminance that depends on time-of-year when processing emotionally charged stimuli. We speculate that it may be through the BMN and medial amygdala that light affects the emotional state in healthy individuals and potentially may also in psychiatric patients, in which different amygdala nuclei may contribute to disorders[25,53-58].

## Methods

This paper arises from a larger study and only describes the methods relevant to the emotional task. Refer to the following papers[44,59-61] for more information about the larger study. The protocol was approved by the Ethics Committee of the Faculty of Medicine at the University of Liège. Participants gave their written informed consent and received monetary compensation. Data acquisitions took place in Liège, Belgium, between December 2020 and August 2023. Additional methodological details are provided as supplementary methods. The processed data supporting the results and analysis script included in this manuscript are publicly available via the following open repository: https://gitlab.uliege.be/CyclotronResearchCentre/Public/fasst/amygdala_7t_light. Access to the raw data is available under request to the corresponding author (GV).

### Participants

Thirty-six healthy participants (23.9 y ± 2.8; 23 women; all Caucasian) took part in the study. They were recruited though a local GDPR complain database of potentially volunteers and through local internet advertisement. Exclusion criteria were assessed through questionnaires and a semi-structured interview and were as follows: history of psychiatric and neurological disorders, sleep disorders, use of psychoactive drugs or addiction; history of ophthalmic disorders or auditory impairments; colour blindness; night shift work during the last year or recent trans-meridian travel during the last 2 months; excessive caffeine (>4 caffeine units/day) or alcohol consumption (>14 alcohol units/week); medication affecting the central nervous system; smoking; pregnancy or breast feeding (women). Their scores on the 21-item Beck Anxiety Inventory[62] and the Beck Depression Inventory-II[30] were minimal or mild (<18) and minimal (<14), respectively. Questionnaires further assessed chronotype with the Horne-Östberg questionnaire[63] and seasonality with the Seasonal Pattern Assessment Questionnaire[31], but the latter two questionnaires were not used for the inclusion of the participants. Participants refrained from caffeinated and alcohol-containing beverages and excessive exercise for at least 3 days before the experiment. The scores on the Beck Depression Inventory-II[30] were used to assess the influence of mood on the regional activity of the amygdala.

Seven datasets were missing or corrupted such that 29 participants (24 y ± 3.1; 18 women; Supplementary Table 1) were included in the present analyses (three participants did not complete the entire task because of scan technical issue interrupting acquisition or subject wanted to exit the scanner before the end of the session, while quality check revealing an important ghosting artefact for 2 datasets and important mismatch in the coregistration of subject space to the MNI space for 2 datasets which could not be resolved following multiple attempts at the time of submitting the manuscript). At least one valid data set was acquired each month of the year and participants are relatively well spread across the 4 seasons (Winter: 4; Spring: 10; Summer: 6; Fall: 9).

### Protocol

Structural brain images were acquired 1 to 2 weeks before the experiment, during a visit which served as habituation to the experimental conditions. Participants then followed a loose sleep-wake schedule (±1 h from habitual sleep/wake-up time) for 7 days, to maintain realistic entrained life conditions and avoid excessive sleep restriction (verified using wrist actigraphy -AX3, Axivity, UK- and sleep diaries). Participants arrived at the laboratory 1.5 after their habitual wake time. To standardise participants' recent light history, they were exposed to 5 min of bright white light (1000 lux; with the chin on a chin-rest, -15 cm away from a plastic diffuser in front of a polychromatic halogen light bulb) and were then maintained in dim light (<10 lux) for 45 min (bright and dim light levels were controlled for each participant at eye level). During the dim light period, participants were given instructions about the fMRI cognitive tasks and completed practice tasks on a luminance-controlled laptop (<10 lux). The fMRI sessions consisted of an executive task (25 min), an attentional task (15 min), and an emotional task (20 min), which is the only task included in the present paper. All participants included in this analysis were therefore scanned -3 h after habitual wake-up time and between 8 am and 1:15 pm [Fig.1A]. Participants always completed the executive task first, as it was the most demanding task. The order of the following two tasks was counterbalanced across participants (45% completed the emotional task before the attentional task). Hence, any bias arising from the relative position of the task could not be truly assessed in our analyses but was controlled for. While in the MR-scanner, participants were asked to keep their eyes open and try not to blink too much during the cognitive tasks. An eye-tracking system (EyeLink 1000Plus, SR Research, Ottawa, Canada) was monitored for proper eye-opening during all data acquisitions. The MRI environment was kept in dim light (<10 lux) during the MRI scan.

### Light exposure

An MRI-compatible light system designed-in-lab was developed to ensure relatively uniform and indirect illumination of participants' eyes whilst in the MRI scanner. An 8-m long MRI-compatible dual-branched optic fibre (1-inch diameter, Setra Systems, MA, USA) transmitted light from a light box (SugarCUBE, Ushio America, CA, USA), that was stored in the MRI control room. The dual end of the optic fibre was attached to a light stand fitted at the back of the MRI coil, allowing for reproducible fixation and orientation of the optic fibre ends. The dual branches illuminated the inner walls of the head coil to indirectly illuminate the participants' eyes. A filter wheel (Spectral Products, AB300, NM, USA) and optical fibre filters to switch between a narrowband 589 nm filter (full width at half maximum: 10 nm) and a UV long-bypass filter (433–1650 nm) filter) and alternate between a monochromatic orange light and the full output of the polychromatic blue-enriched LEDs (6500 K) (Fig. 1B and Supplementary Table S2 for in-detail light characteristics). The spectra of the lights were assessed at the level of the end of the optic fibre (AvaSpec-2048, Avantes, The Netherlands). Illuminance could not be measured directly in the magnet, but the light source was calibrated prior to the experiment (840-C power meter, Newport, Irvine, CA).

Blue-enriched light illuminances were set according to the technical characteristics of the light source and to keep the overall photon flux similar to prior 3T MRI studies of our team using the same emotional task (between -$10^{12}$ and $10^{14}$ ph/cm²/s)[21]. The orange light was introduced as a control visual stimulation for potential secondary whole-brain analyses. For the present region of interest analyses, we discarded colour differences between the light conditions and only considered illuminance as indexed by mel EDI lux, constituting a limitation of our study.

## Emotional task

The task consisted of gender discrimination of auditory vocalisations [Fig. 2C][64]. Participants were asked to use a keypad to indicate what they believed the gender of the person pronouncing each token was. The gender classification was a lure task: its purpose was to trigger an emotional response, as participants were not told that 50% of the stimuli were pronounced with angry prosodies. The 240 auditory stimuli were pronounced by professional actors (50% women) and consisted of three meaningless words ("*goster*", "*niuwenci*", "*figotleich*"). The stimuli were expressed in either an angry or neutral prosody (validated by behavioural assessments[64] and in previous experiments[28,29]). During each 30–40-s light block, four angry prosody stimuli and four neutral prosody stimuli were presented in a pseudorandom order and delivered every 3–5 s. A total of 160 distinct voice stimuli (50% angry; 50% neutral) were distributed across the four light conditions. The darkness periods separating light blocks contained two angry and two neutral stimuli (80 stimuli in total). The emotional task (together with light exposure changes) was programmed with Opensesame (3.2.8)[65]. Participants heard the auditory stimuli through MR-compatible headphones (Sensimetrics, Malden, MA) and the volume was set by the participant before starting the tasks to ensure a good auditory perception. Participants used an MRI-compatible keypad to respond to task items (Current Designs, Philadelphia, PA). The instruction was to privilege accuracy over rapidity when responding. The auditory stimuli used in the task were matched for the duration (750 ms) and mean acoustic energy to avoid loudness effects.

## Data acquisition

MRI data were acquired in a 7T MAGNETOM Terra MR (Siemens Healthineers, Erlangen, Germany) with a 32-channel receive and 1-channel transmit head coil (Nova Medical, Wilmington, MA, USA). Dielectric pads (Multiwave Imaging, Marseille, France) were placed between the subject's head and receiver coil to homogenize the magnetic field of Radio Frequency (RF) pulses. The multi-band Gradient-Recalled Echo - Echo-Planar Imaging (GRE-EPI) sequence with axial slice orientation was set as follows: TR = 2340 ms, TE = 24 ms, FA = 90°, no interslice gap, in-plane FoV = 224 mm × 224 mm, matrix size = 160 × 160 × 86, voxel size = (1.4 × 1.4 × 1.4) mm³). To avoid saturation effects, the first three scans were discarded. To correct for physiological noise in the fMRI data the participants' pulse and respiration movements were recorded using a pulse oximeter and a breathing belt (Siemens Healthineers). Following the fMRI acquisition a 2D GRE field mapping sequence to assess B0 magnetic field inhomogeneities with the following parameters: TR = 5.2 ms, TEs = 2.26 ms and 3.28 ms, FA = 15°, bandwidth = 737 Hz/pixel, matrix size = 96 × 128, 96 axial slices, voxel size = (2 × 2 × 2) mm³, acquisition time = 1:38 min. The Magnetization-Prepared with 2 RApid Gradient Echoes (MP2RAGE) sequence was set as follows: TR = 4300 ms, TE = 1.98 ms, FA = 5°/6°, TI = 940 ms/2830 ms, bandwidth = 240 Hz, matrix size = 256 × 256, 224 axial slices, acceleration factor = 3, voxel size = (0.75 × 0.75 × 0.75) mm³.

## Data pre-processing

For the MP2RAGE images, the background noise was removed using an extension (https://github.com/benoitberanger/mp2rage) of Statistical Parametric Mapping 12 (SPM12; https://www.fil.ion.ucl.ac.uk/spm/software/spm12/) under Matlab R2019 (MathWorks, Natick, Massachusetts)[66]. Then the images were reoriented using the 'spm_auto_reorient' function (https://github.com/CyclotronResearchCentre/spm_auto_reorient) and corrected for intensity non-uniformity using the bias correction method implemented in the SPM12 "unified segmentation" tool[67]. To ensure optimal co-registration, brain extraction was done using SynthStrip[68] in Freesurfer (http://surfer.nmr.mgh.harvard.edu/). The brain-extracted T1-images were used to create a T1-weighted group template using Advanced Normalization Tools

(ANTs, http://stnava.github.io/ANTs/) before normalization to the Montreal Neurological Institute (MNI) space using ANTs (1 mm³ voxel; MNI 152 template).

For the EPI images, auto reorientation was applied on the images first. Then, voxel-displacement maps were computed from the phase and magnitude images associated with B0 map acquisition, using the SPM fieldmap toolbox. To correct for head motion and static and dynamic susceptibility-induced variance, "Realign & Unwarp" of SPM12 was then applied to the EPI images. The realigned and distortion-corrected EPI images then underwent brain extraction using the SynthStrip and then the final images were smoothed with a Gaussian kernel characterised by a full width at half maximum of 3 mm.

## Data analysis

The brain-extracted anatomical T1-images were used to create a T1-weighted group template using Advanced Normalization Tools (ANTs, http://stnava.github.io/ANTs/) before normalization to the Montreal Neurological Institute (MNI) space using ANTs (1 mm³ voxel; MNI 152 template). For each subject, the first level analysis of fMRI data was performed in the native space. Before the group analysis, statistical maps were first transferred to the group template space and then the MNI space (1 × 1 × 1 mm³ resolution; MNI 152 template) using ANTs. The subject level GLM included a high-pass filter with a 256 s cut-off to remove low-frequency drifts as well as movement and physiological parameters (cardiac, and respiration), which were computed with the PhysIO Toolbox (Translational Neuromodeling Unit, ETH Zurich, Switzerland)[69]. All these additional regressors were included as covariates of no interest.

Our statistical analyses consisted of an a priori region of interest focused on the activity of the amygdala which was estimated as part of a whole brain general linear mixed model computed with SPM12. The auditory stimuli were modelled as stick functions convolved with a canonical hemodynamic response function. There were two parts to the whole-brain analysis. In the main analysis, we assessed brain responses during the task and how they were modulated by overall changes in illuminance level. The two regressors of task events (neutral, angry) were each accompanied by a parametric modulation corresponding to the light melanopic illuminance level (0, 0.16, 37, 92, 190 mel EDI). The contrasts of interest consisted of the main effect of the task (emotional vs. neutral stimuli) and their parametric modulations (emotional vs. neutral stimuli x illuminance). In the next whole-brain analysis, we assessed the responses to the stimuli under each light condition. Separate regressors modelled each task's event type under each light condition. The contrasts of interest consisted of the main effects of each regressor. Whole-brain group results over the entire sample were used for visualisation purposes only.

We used an amygdala atlas to segment the region into 10 subparts (bihemispheric) in the MNI subject space[32], corresponding to nuclei or nucleus groups (Fig. 2.A). The REX Toolbox (https://web.mit.edu/swg/software.htm) was used to extract the activity estimates (betas) from contrast of interest in each amygdala subpart[70]. In the first analysis, this yielded 1 activity estimate per contrast, per stimulus type and per amygdala subpart and in the second analysis, we obtained 5 activity estimates per stimulus and subpart.

## Statistics

Statistical analyses of the activity of each subpart were computed in SAS 9.4 (SAS Institute, NC, USA) and consisted of (2-sided) Generalised Linear Mixed Models (GLMM) with the subject as a random factor (intercept and slope) and were adjusted for the dependent variable distribution. As main the statistical analysis included all subparts, light conditions and stimulus types in a single model (when relevant), the significance threshold was not corrected for multiple comparisons and was set at $p < 0.05$. Direct post hoc of the main analyses were corrected for multiple comparisons using a simulated adjustment. Activity

estimates were considered outliers if > ±3 SDs across emotional stimuli and light level and were removed.

The two main analyses included the average activity estimates to emotional and neutral stimuli i) independent of light and ii) modulated by illuminance as a dependent variable and the amygdala subparts and stimulus type (neutral/angry; as repeated measure with compound symmetry correlation), together with their interaction and age and sex as covariates. BMI was also part of the covariates because it could influence brain activity, including in regions involved in reward, which is related to emotional processing[71]. The first main analysis indicated that 3 areas did not respond to the task, these were therefore discarded from all further analyses related to time of year, mood and light acute impact.

To test for variations related to time of year and mood, both models were recomputed respectively with a time of year covariate consisting of the cosine value of the day of the year expressed in degrees of the 360-day year (365 day = 360°; 1 day = 0.986°; December 21st = 0°) and the score of Beck Depression Inventory-II[30]. Three-way interaction terms were included in the model [subpart x time of year x mood] as well as the three simple interactions composing the three-way interaction.

The main analysis focusing on light exposure impact used a linear parametric modulation of the changes in activity with increasing illuminance to grasp the overall impact of illuminance change. Since the latter may have missed non-linear changes in activity with changes in illuminance, a second GLMM included the four amygdala subparts that were showing an impact of overall illuminance change. The GLMM included the activity estimates of the four subparts as the dependent variable and amygdala subpart, stimulus type (neutral/angry; as repeated measure with compound symmetry correlation) and illuminance (0, 0.16, 37, 92, 190 mel EDI lux; as the repeated measures nested within stimulus type - compound symmetry correlation), together with age, sex and BMI as covariates. A three-way interaction term was included in the model [subpart x illuminance x stimulus type] as well as the three simple interactions composing the three-way interaction.

A final set of exploratory GLMMs reported in the discussion section included performance metrics as dependent variables (reaction time – ms) and included the activity of each of the four amygdala subparts that were showing an impact of overall illuminance change in separated model together with stimulus type and illuminance (repeated measures nested within stimulus type - compound symmetry correlation), together with age, sex and BMI as covariates.

Optimal sensitivity and power analyses in GLMMs remain under investigation (e.g. ref. 72). We nevertheless computed a prior sensitivity analysis to get an indication of the minimum detectable effect size in our main analyses, given our sample size. According to G*Power 3 (version 3.1.9.7)[73], taking into account a power of 0.8, an error rate α of 0.05, and a sample of 29 allowed us to detect large effect sizes $r > 0.53$ (two-sided; absolute values; CI: 0.2–0.75; $R^2 > 0.28$, $R^2$ CI: 0.04–0.56) within a linear multiple regression framework including two tested predictors (illuminance effect, amygdala subpart) and up to six covariates (stimulus type, age, sex, BMI, season and affective status).

### Reporting summary
Further information on research design is available in the Nature Portfolio Reporting Summary linked to this article.

## Data availability
Source data are provided with this paper. The processed data supporting the results included in this manuscript are publicly available via the following open repository: https://gitlab.uliege.be/CyclotronResearchCentre/Public/fasst/amygdala_7t_light. The raw data is available under restricted access because it could be identified and linked to a single subject and represent a large amount of data.

Access can be obtained by sending a request to the corresponding author (GV), who will respond within 1 month. Data sharing will require evaluation of the request by the local Research Ethics Board, and approval of the submitted request usually take two months. A data transfer agreement (DTA) will need to be signed following ethics approval. Source data are provided with this paper.

## Code availability
The analysis scripts supporting the results included in this manuscript are publicly available via the following open repository:

https://gitlab.uliege.be/CyclotronResearchCentre/Public/fasst/amygdala_7t_light.

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

## Acknowledgements

The study was conducted at the GIGA-In Vivo Imaging technological platform of ULiège, Belgium. The authors thank Christine Bastin, Annick Claes, Alexandre Berger, Christian Degueldre, Catherine Hagelstein, Gregory Hammad, Brigitte Herbillon, Patrick Hawotte, Ekaterine Koshmanova, Sophie Laloux, Erik Lambot, Benjamin Lauricella, André Luxen, Pierre Maquet, and Eric Salmon for their help over the different steps of the study. It was supported by the Belgian Fonds National de la Recherche Scientifique (FNRS; CDR J.0222.20, & J.0216.24), the European Union's Horizon 2020 research and innovation program under the Marie Skłodowska-Curie grant agreement No 860613, the Fondation Léon Frédéricq, ULiège - U. Maastricht Imaging Valley, ULiège-Valeo Innovation Chair "Health and Well-Being in Transport" and Sanfran (LIGHT-CABIN), Fondation Recherche Alzheimer (SAO-FRA 2019/0025 & 2022/0014), the European Regional Development Fund (Biomed-Hub, WALBIOIMAGING), and Siemens. None of these funding sources had any impact on the design of the study nor the interpretation of the findings. E.B. was supported by the Maastricht University - Liège University Imaging Valley. R.S. and F.B. were supported by the European Union's Horizon 2020 research and innovation program under the Marie Skłodowska-Curie grant agreement No 860613. I.C., I.P., N.M., and G.V. are/were supported by the FRS-FNRS. P.T. and L.L. are/were supported by the EU Joint Programme Neurodegenerative Disease Research (JPND) IRONSLEEP and SCAIFIELD projects, respectively – FNRS references: PINT-MULTI R.8011.21 & R.8006.20). L.L. is supported by the European Regional Development Fund (WALBIOIMAGING). M.Z. is supported the Fondation Recherche Alzheimer (SAO-FRA 2022/0014).

## Author contributions

Study concept and design by I.C. and G.V. Data acquisition and analysis by I.C., J.F.B.A., R.S., N.M., P.T., E.B., & I.P. Methodological support and/or support in interpreting the data by C.P., F.C., M.Z., L.L. Funding was mostly obtained by F.C., C.P., and G.V. I.C. and G.V. drafted the first version of the manuscript. All authors revised the manuscript and had final responsibility for the decision to submit for publication.

## Competing interests

The authors declare no competing interests.
