## [Transparent Peer Review file · Nature Communications]

Regional activity within the human amygdala varies with season, mood and illuminance

Corresponding Author: Professor Gilles Vandewalle

Version 0:

Reviewer comments:

Reviewer #1

(Remarks to the Author)

In this study, the authors used 7 Tesla fMRI to test whether activation level in the human amygdala during emotional vs. neutral auditory stimuli vary with time of year, affective state, and illuminance level. The use of a strong magnetic field enabled characterizing the effects in 10 subregions of the amygdala.

The sample of 29 participants allowed the detection of medium effect sizes, and thus, it is adequate. While the authors controlled for sex and age, the age range was quite narrow ($24y \pm 3.1$) and men were underrepresented (18 women, 11 men). Additionally, the distribution of participants across seasons was not uniform (Winter, 4; Spring, 10; Summer, 6; Fall: 9). Collecting data from a larger age range and from equal representation of women and men would have been ideal, but considering the difficulty and complexity of the experiment, such limitations are well accepted in the field.

The study shows that activity in subregions 3 (basomedial nucleus) and 9 (anterior amygdaloid area) increased as day length decreased, while activity in subregions 5 (medial and cortical nuclei) and 7 (amygdala transition area) increased as affective state deteriorated. Thus, activity in selected nuclei of the amygdala is either related to season or affective state. Activity in the four subparts that varied with season or affective state (3, 5, 7, and 9), negatively correlated with acutely-modulated illuminance level, when processing emotional stimuli but not neutral stimuli. Overall, these findings the possibility that photoperiod and acute light exposure might influence emotion through modulation of selected amygdalar nuclei. As such, the study could pave the way toward better understanding of the benefits of light therapy in the treatment of affective disorders.

Major comments:

Figure 1 – Since the order of the emotional and attentional tasks was random, how did the relative position of the emotional task interact with the effect of time of year, affective state and illuminance on amygdala activation?

Line 162 – It is not clear how df of ~500 was achieved. It would be useful to describe the number of blocks and repetitions that led to the df. The sampling unit should probably be the number of participants (29 people), and thus df should not exceed 500. If this expectation is wrong, please explain why. This comment applies to all other instances where similarly large df is reported.

Figures 1 and 2 – Considering the typically modest variation of activity with of season, affective state, and illuminance level, it would be useful to specify the effect size of the main effect GLMM and post-hoc tests.

The manuscript would benefit tremendously from thorough editing.

Minor comments:

Title – Would probably benefit from being more informative. For example, “Activity in selective human amygdala subregions varies with season, emotion, and illuminance”.

Line 48 – “by the reduced shorter photoperiod” – not clear.

Line 108 – “in the morning” – Specify the time range when fMRI was performed.

Line 109 – It is not clear why only the ‘emotional task’ out of the three tasks performed was analyzed?

Line 112 – The three illuminance levels (37, 92, 190 melanopic EDI) are low compared to about 10,000 lux melanopic EDI or more around noon outdoors. Since these low levels induced changes in amygdala activation, daylight would likely evoke much larger changes.

Figure 1, caption – The author should state that the ‘prior light history standardization’ consisted of 5-min 1000 lux light and the following 45-min under dim light.

Figure 1H,I – The authors should describe how the sine function was fitted to data, and report measures of the goodness of fit.

Line 155 – It is not clear why BMI should be controlled for. Is there any evidence for variation in brain activation according to BMI?

Line 216 – Euro sign instead of (E).

Figure 2, panels F-M – It would be useful to indicate the subpart number on each figure panel.

Figure 2, Panels H-I – In addition to ‘day of the year’ it would be useful to include another x-axis indicating the months. This will make it easier to appreciate the claim in line 185 “responses were stronger during winter”.

Suppl. Table S4 – This table is very large, comprising multiple sections. As the table is being referred to many times, it is hard to identify the relevant table section. Either refer to specific sections of the table or divide the table into several smaller tables.

Lines 185-188 – Fig. 2A is referenced but, if I understand correctly, the results appear in panels J and K.

Lines 193-194 – “Overall, the results suggest that the activity of the nuclei composing the medial and superior region of the amygdala (encompassing 4 to 5 subparts of our template) is related to season and/or affective state.” This claim is not supported by the results. A more accurate claim would be: “Activity in subparts 3 and 9 varied significantly with season, while activity in subparts 5 and 7 varied significantly with affective state. Thus, activity in selected nuclei comprising the amygdala’s medial and superior region is either related to season or affective state”.

Line 300 – Again, the conclusions should be stated more carefully. “We found that the activity of medial and superior nuclei of the amygdala was related...” should be “We found that the activity in selected medial and superior nuclei of the amygdala was related...”.

Lines 393-395 – The authors correctly suggest that “The cross-talk with the prefrontal cortex is also of interest given the changes in functional connectivity previously reported following light therapy when considering the amygdala as whole.” In addition, given the 2022 report of the effect of light on activity in the human prefrontal cortex, at least some of the illuminance-evoked activity modulations observed in the amygdala might be driven by input from the prefrontal cortex, and vice versa, some of the sensitivity observed in the prefrontal cortex might reflect input from the amygdala. This idea is supported by the evidence of effect of light on prefrontal cortex-amygdala connectivity, as indicated in the Introduction section. The Discussion section is well written, addressing the main results, specifying limitations, and raising interesting implications. It could probably be shortened, though.

Lines 663-670 – In the description of prior sensitivity analysis, there is no mention of the predictors: seasons and affective state.

Reviewer #2

(Remarks to the Author)

The authors present a study investigating the impacts of varying light intensities, season, and affective state (depressive symptoms) on regional activity in the amygdala using high resolution MRI. The study is well presented, and the findings interesting. However, they are at times overstated given the relatively weak associations observed. Additionally, some aspects of the method could be clearer. Specific comments below:

Abstract:

1. The authors could be more specific here about the continuous nature of assessments across time of year (as opposed to distinct groups in each season)
2. The results could be clearer as to the direction of the affect/season related results i.e., “the impact of light on part of these nuclei varied with season and affective state” could be more specific.

Introduction:

3. Minor typo/grammatical error “reduced shorter photoperiod” is repetitive

4. The paragraph starting at line 77 is a little hard to follow. Ref 21 found increased amygdala-prefrontal connectivity during light exposure, not decreased (aligned with findings from ref 22 post light therapy). It would be clearer to present the primary findings of the four studies described here before speculating on potential reasons for apparent differences (i.e., suppression of activity by light during rest, and reduced activity following light therapy, but not during the completion of an emotional task). It could be clearer why the presence of an emotional task would explain these differences, though this may be more appropriate for the discussion where it can be explored in the context of the current findings.

Results:

5. Although full methods are presented later, it would be helpful to specify the time frame of 'morning' here.

6. Minor typo in the caption of Figure 1 – 'lose' sleep-wake schedule (loose?). The prior light history standardisation is also not clear, the text sounds more like it is light exposure across those 7 days, when in fact it is only the immediate light history prior to the scan.

7. It would be helpful to indicate local season on the graphs which show data by day of year (and/or actual changes in photoperiod).

8. The way in which affective state was measured should be noted here.

9. Many of the associations shown in the bottom section of Figure 2 were very weak. Given that some did not survive correction, the authors should show/state this more clearly on the figure.

Discussion:

10. "Affective state varies with seasons in a large portion of the healthy population" – given depression scores (the measure of 'affective state' used in this study) did not vary by season in this sample, there may be a better way to introduce the summary of your findings.

11. The discussion overall is well reasoned and written. However, at times the results, particularly those pertaining to affective state and season are overstated given the relatively weak relationships presented.

Methods:

12. Minor typo/clarification needed re: "Subject willing to exit the scanner" – perhaps supposed to be wishing to exit?

13. Additional information regarding the standardisation of prior light would be helpful – i.e., are these measures the maximal/minimal level in the room, how was 1000 lux delivered, using a light therapy device? Additional descriptions of the geometric space would be helpful to understand the actual retinal light exposure. Additionally, specification of the lighting in the MRI room should be included.

14. Can the authors speak to their decision to use the BDI as a marker of 'affective state'? They also had BAI available, which is arguably also relevant to the regions of interest, and appears to show a greater range of scores, but another measure which may be more sensitive to normal mood variation in healthy persons would have likely been more appropriate than either. I am concerned that the limited range of BDI scores may not meaningfully reflect differences in 'affective state' in this population. It would be more appropriate to refer to this measure as depressive symptoms, rather than affective state, throughout the paper. At a minimum the actual range of scores and mean/SD should be reported here in the methods, but ideally the limitations of using this measure in an otherwise healthy population should also be discussed in the paper.

Version 1:

Reviewer comments:

Reviewer #1

(Remarks to the Author)

The authors have addressed my concerns, and the revised manuscript is significantly improved. I have no further comments.

Reviewer #2

(Remarks to the Author)

The authors have thoughtfully revised their manuscript. One minor note regarding the new figure panel (Fig 3f) - the embedded figure does not appear to be the new version. I would also suggest adding the local season "names" to the figure in addition to colours for ease of interpretation.

Reviewer #1 (Remarks to the Author):

The sample of 29 participants allowed the detection of medium effect sizes, and thus, it is adequate. While the authors controlled for sex and age, the age range was quite narrow ($24y \pm 3.1$) and men were underrepresented (18 women, 11 men). Additionally, the distribution of participants across seasons was not uniform (Winter, 4; Spring, 10; Summer, 6; Fall: 9). Collecting data from a larger age range and from equal representation of women and men would have been ideal, but considering the difficulty and complexity of the experiment, such limitations are well accepted in the field.

Overall, these findings the possibility that photoperiod and acute light exposure might influence emotion through modulation of selected amygdalar nuclei. As such, the study could pave the way toward better understanding of the benefits of light therapy in the treatment of affective disorders.

We would like to thank Reviewer 1 for their detailed evaluation of our manuscript and for highlighting the strengths and limitations of the research as well as underlying its interest. We believe the comments helped improving significantly the quality of our paper.

Major comments:

- Figure 1 – Since the order of the emotional and attentional tasks was random, how did the relative position of the emotional task interact with the effect of time of year, affective state and illuminance on amygdala activation?

We cannot truly assess how the relative position of the emotional task interacts with the effect of time-of-year, mood* and illuminance as this would mean adding a factor in the analyses and splitting the sample in two. However, as indicated in the method section (page 22/23), the order of the emotional and attentional task was counterbalanced across participants so that approximately half of the sample complete the emotional first and the other half second (% completed the emotional task before the attentional task). Hence, any potential bias associated to the relative positive was controlled for.

We added this aspect to the text (line 523-527: *'Participants always completed the executive task first as it was the most demanding task. The order of the following two tasks was counterbalanced across participants (45% completed the emotional task before the attentional task). Hence, any bias arising from the relative position of the task could not be truly assessed in our analyses but was controlled for.'*

* Please note, that following a comment of reviewer 2, we changed the terms “affective state” to “mood”.

- Line 162 – It is not clear how df of ~500 was achieved. It would be useful to describe the number of blocks and repetitions that led to the df. The sampling unit should probably be the number of participants (29 people), and thus df should not exceed 500. If this expectation is wrong, please explain why. This comment applies to all other instances where similarly large df is reported.

We report both the standard DF and those associated with the error term, the latter depending on the total number of observations (minus the number of model parameters). Having 29 subjects x 10 amygdala subparts x 2 task conditions can therefore lead to error DF close to 500 following Kenward-Roger estimation.

- Figures 1 and 2 – Considering the typically modest variation of activity with of season, affective state, and illuminance level, it would be useful to specify the effect size of the main effect GLMM and post-hoc tests.

Partial effect size (R^{2*}) of the significant association are/were reported in the main text and in the supplementary tables.

- The manuscript would benefit tremendously from thorough editing.

We have gone through and edited the manuscript and hope to have made it clearer.

Minor comments:

- Title – Would probably benefit from being more informative. For example, “Activity in selective human amygdala subregions varies with season, emotion, and illuminance”.

We thank the reviewer for their suggestion. We modified the title to “***Regional activity within the human amygdala varies with season, mood and illuminance***”.

- Line 48 – “by the reduced shorter photoperiod” – not clear.

We have modified the text accordingly, ‘*driven in part by the shorter photoperiod taking place during the fall and winter seasons*’ (line 53).

- Line 108 – “in the morning” – Specify the time range when fMRI was performed.

We now specify the following:

*'The brain activity of 29 healthy young participants (24y ±3.1; 18 women) was recorded during an fMRI protocol conducted in Liège (Belgium) in the morning (**scans 3 to 3.5hrs after habitual wake-up time**),...'* (line 119-120);

*'**Participants arrived at the laboratory 1.5h after wake-up time**'* (line 129 – figure 1 caption & line 513 - methods);

*'In total, 29 participants (24y ±3.1; 18 women) performed an executive (always first), an emotional and an attentional task (pseudo-randomly 2nd or 3rd, purple arrow; **meaning that the fMRI recording of the emotional task was completed 3 to 3.5h after wake-up time**).' (line 133-134, caption figure 1);*

*'**All participants included in this analysis were therefore scanned ~3h after habitual wake-up time and between 8 am and 1:15 pm [Fig.1A].'*** (line 521-523, methods)

- Line 109 – It is not clear why only the 'emotional task' out of the three tasks performed was analyzed?

The emotional task was the sole focus of the paper as we wanted to investigate the impact of light, season, and mood, on emotional processing in the amygdala, which is a key region for this aspect of brain function. In addition, the gender discrimination of auditory vocalisations task we used has previously been shown to show that the responses of the amygdala during the task were influenced by light^{1,2}. We have previously analysed the executive and attentional task (see the following papers³⁻⁶). We have added the following to the text:

*'We took advantage of the higher resolution of ultra-high-field (UHF) 7 Tesla (7T) fMRI to record the activity of the amygdala in healthy young adults (**devoid of psychiatric disorders**) exposed to the light of various illuminances while engaged in an auditory emotional task **which was previously successfully used to show an influence of light on the activity of the amygdala**'.* line 105-109.

*'This paper arises from a larger study and only describes the methods relevant to the emotional task. **A full description of the methods has been previously described.**'* (line 476-477).

- Line 112 – The three illuminance levels (37, 92, 190 melanopic EDI) are low compared to about 10,000 lux melanopic EDI or more around noon outdoors. Since these low levels induced changes in amygdala activation, daylight would likely evoke much larger changes.

The blue-enriched light levels for the study were set according to the technical characteristics of the light source, to avoid too much glare effect following the adaptation

to dim light, and to keep the overall photon flux similar to prior 3T MRI studies of our team (between $\sim 10^{12}$ and 10^{14} ph/cm²/s)^{7,8}. The orange light was introduced as a control visual stimulation for potential secondary whole-brain analyses. For the present region of interest analyses, we discarded colour differences between the light conditions and only considered illuminance as indexed by mel EDI lux. As indicated in the text, this constitutes a limitation of our study as it does not allow attributing the findings to a particular photoreceptor class. Higher light levels, such as 10,000 lux melanopic EDI or more around noon outdoors, could evoke larger changes, although photoreceptor adaptation (driven in part by ipRGCs) may mitigate the change in absolute illuminance. The text has been modified as follows to take the latter aspect into account (line 456-460): “*While compatible with usual indoor levels, the light we administered was far from outdoor levels, and we can only speculate that the effects we report are similar under these outdoor conditions. The effects of outdoor light conditions may be stronger, but the light adaptation processes of the retina, in part driven by ipRGCs, may mitigate differences in absolute illuminance*⁹.”

- Figure 1, caption – The author should state that the ‘prior light history standardization’ consisted of 5-min 1000 lux light and the following 45-min under dim light.

We have edited the figure 1 caption and included the following ‘*The prior light history of participants was standardised **on the morning of the MRI scan**, consisting of 5-min 1000 lux light and the following 45-min under dim (<10 lux) light.*’ (line 130).

- Figure 1H, I – The authors should describe how the sine function was fitted to data, and report measures of the goodness of fit.

Statistical analyses consisted of Generalised Linear Mixed Models (GLMM) which included a time-of-year covariate consisting of the cosine value of the day of the year expressed in degrees of the 365-day year (365 day = 360°; 1 day = .986°; December 21st = 0°). For display purposes only, the figure included a cosine function fit to the data which does not replace the outcomes of the GLMM so that its goodness of fit is not relevant to our analyses.

We added the following to the text regarding this aspect:

‘Reaction times were not significantly affected by illuminance (main effect, illuminance; $F_{4,221} = .92, p = .45$) or by time-of-year (using the cosine value associated with day of the year over a year-long cycle, see methods; main effect, time-of-year; $F_{1,25} = .12, p = .73$).’ (line 158-159)

‘Cosine fits are included for display purposes only and do not replace the outcomes of the GLMMs.’ (line 233 – caption of figure 2)

- Line 155 – It is not clear why BMI should be controlled for. Is there any evidence for variation in brain activation according to BMI?

BMI was included as a reflection of overall health status and because previous studies using fMRI reported brain responses and brain structure could be influenced by particularly in individuals with higher BMI (which we excluded), including in regions related to reward, an aspect that is related to emotional processing. Repeating the analyses without BMI as a covariate does not influence the statistical outcome of our analyses. We updated the text regarding this aspect as follows (line 657-658): *‘ BMI was also part of the covariates because it could influence brain activity, including in regions involved in reward, which is related to emotional processing¹⁰.’*

- Line 216 – Euro sign instead of (E).

Response: We have edited the text accordingly (line 227).

- Figure 2, panels F-M – It would be useful to indicate the subpart number on each figure panel.

We have edited the Figure 2 so that the subpart number and name is included as a title for each panel. (See Figure 2 – F-M)

- Figure 2, Panels H-I – In addition to ‘day of the year’ it would be useful to include another x-axis indicating the months. This will make it easier to appreciate the claim in line 185 “responses were stronger during winter”.

We have updated the figure to include a coloured panel that shows the changing of the seasons (Spring, Summer, Autumn, Winter). See Figure 2.H-I.

- Suppl. Table S4 – This table is very large, comprising multiple sections. As the table is being referred to many times, it is hard to identify the relevant table section. Either refer to specific sections of the table or divide the table into several smaller tables.

We have added section headers to supplementary table S4 (**a to e**) so it is easier for readers to follow. We have updated the main manuscript to reflect this.

- Lines 185-188 – Fig. 2A is referenced but if I understand correctly, the results appear in panels J and K.

We have edited the figure labelling in text so it is clearer that we are referring to Fig. 2A to orient the reader about which subpart we are discussing. The figure results are referenced later in the sentence. “*Post hoc tests then revealed worse affective state was significantly associated ... (t= 4.28, $p_{corrected}$ = .0002, subpart 5, violet on Fig. 2A) and amygdala transition areas (ATA, t= 4.64, $p_{corrected}$ <.0001, subpart 7, orange on Fig. 2A) (both subparts were also different from part of the other subparts in their association with time-of-year) (Fig. 2J-K, Suppl. Table S4).*” Lines 202, 203, 205.

Additionally, we have corrected the following figure labelling in the text (lines 173 & 179): “*...task was successful in triggering emotional responses in the amygdala with higher response to emotional than to neutral stimuli (GLMM; main effect, stimulus type, $F_{1, 27.74}$ = 9.93; p = .0039, R^2 = .26; Fig. 2D) (Suppl. Table S3). ... visualization, responses significantly differed across amygdala subparts (GLMM; main effect, subpart, $F_{9, 499.9}$ = 2.13; p = .0256, R^2 = .04; Fig. 2E)*”.

- Lines 193-194 – “Overall, the results suggest that the activity of the nuclei composing the medial and superior region of the amygdala (encompassing 4 to 5 subparts of our template) is related to season and/or affective state.” This claim is not supported by the results. A more accurate claim would be: “Activity in subparts 3 and 9 varied significantly with season, while activity in subparts 5 and 7 varied significantly with affective state. Thus, activity in selected nuclei comprising the amygdala’s medial and superior region is either related to season or affective state”.

We have changed this so it is more accurately stated to “*Overall, the results suggest the activity in subparts 3 and 9 varied significantly with the time of the year, while activity in subparts 5 and 7 varied significantly with mood. Thus, activity in selected nuclei comprising the amygdala’s medial and superior region is either related to time-of-year or mood.*” (line 208-211).

- Line 300 – Again, the conclusions should be stated more carefully. “We found that the activity of medial and superior nuclei of the amygdala was related...” should be “We found that the activity in selected medial and superior nuclei of the amygdala was related...”.

We have changed the wording so that it is clearer to the reader. ‘*We found that the activity in selected medial and superior nuclei of the amygdala was related to either time-of-year or mood.*’ (line 319).

- Lines 393-395 – The authors correctly suggest that “The cross-talk with the prefrontal cortex is also of interest given the changes in functional connectivity previously reported following light therapy when considering the amygdala as whole.” In addition, given the 2022 report of the effect of light on activity in the human prefrontal cortex, at least some of the illuminance-evoked activity modulations observed in the amygdala might be driven by input from the prefrontal cortex, and vice versa, some of the sensitivity observed in the prefrontal cortex might reflect input from the amygdala. This idea is supported by the evidence of effect of light on prefrontal cortex-amygdala connectivity, as indicated in the Introduction section.

We modified the discussion as follows to take the comment into account (line 405): “*Ultimately, although we favour an impact of light on the amygdala through a **direct or indirect** projection of the retina to the medial nuclei, all four subparts could equally contribute to the decrease BOLD signal we detect under higher illuminance.¹¹ **Given the changes in functional connectivity previously reported following light therapy when considering the amygdala as a whole,¹² part of our results may be mediated by an impact of light on the prefrontal cortex or on cross-talk with the prefrontal cortex. In addition, our findings support that the reduced responsiveness of (part of) the amygdala does not require a week of light therapy and is already present acutely, during the exposure. The sizes of the effects we detected are small, such that light exposure may be beneficial to the emotional state through a repeated effect on the BMN and medial parts of the amygdala (or on the prefrontal cortex). Again, complex connectivity studies considering light exposure at different timescales are required to test these hypotheses and to assess how our findings fit within a larger network of (most often) small brain regions.**^{13,14}*” lines 406-418

- The Discussion section is well written, addressing the main results, specifying limitations, and raising interesting implications. It could probably be shortened, though.

We have gone through the discussion and edited it, so it is shorter and more concise.

- Lines 663-670 – In the description of prior sensitivity analysis, there is no mention of the predictors: seasons and affective state.

We had initially computed the sensitivity analyses only for our first analyses which included 4 covariates. We added season and affective state, which together with the

correction of an initial error, yields a sensitivity for large effect size of $r > .53$ and $R^2 > .28$ (instead of $r > .47$ and $R^2 > .22$).

Reviewer #2 (Remarks to the Author):

The authors present a study investigating the impacts of varying light intensities, season, and affective state (depressive symptoms) on regional activity in the amygdala using high resolution MRI. The study is well presented, and the findings interesting. However, they are at times overstated given the relatively weak associations observed. Additionally, some aspects of the method could be clearer. Specific comments below:

Response: We would like to thank Reviewer 2 for their thoughtful and detailed evaluation of our work, which helped, in our view, improve the text.

Abstract:

1. The authors could be more specific here about the continuous nature of assessments across time of year (as opposed to distinct groups in each season)

We edited the abstract accordingly.

*“We first **considered time-of-year** changes in activity that are related to the slow change in photoperiod. We find that the response to emotional stimuli of **selected medial and superior nuclei of the amygdala peaked around the start of winter or increased with worse mood status**. We further assessed how alternating short exposures to light of different illuminance acutely affected the regional activity of the amygdala....”* Line 38 & 42.

Additionally, we have edited the manuscript so it is clear we are measuring seasonality in terms of days of the year. We have used ‘time-of-year’ instead of ‘season’, so it is clear about the continuous nature of the time-of-year assessment.

2. The results could be clearer as to the direction of the affect/season related results i.e., “the impact of light on part of these nuclei varied with season and affective state” could be more specific.

We have edited the results section so it is more specific: *“We find that the activity in selected medial and superior nuclei of the amygdala **peaked around the start of winter or increased with worse mood status**”* (lines 40-41) and *“Importantly also, **the impact of light on part of these nuclei peaked around the start of summer or decreased with worse mood.**”* (line 45-46)

Introduction:

3. Minor typo/grammatical error “reduced shorter photoperiod” is repetitive

We have fixed the manuscript accordingly (line 48).

4. The paragraph starting at line 77 is a little hard to follow. Ref 21 found increased amygdala-prefrontal connectivity during light exposure, not decreased (aligned with findings from ref 22 post light therapy). It would be clearer to present the primary findings of the four studies described here before speculating on potential reasons for apparent differences (i.e., suppression of activity by light during rest, and reduced activity following light therapy, but not during the completion of an emotional task). It could be clearer why the presence of an emotional task would explain these differences, though this may be more appropriate for the discussion where it can be explored in the context of the current findings.

We have re-written this section to clarify these aspects while still keeping the introduction concise (Lines 81 to 97): ***“In vivo neuroimaging studies on the topic in Humans are scarce and have reported contrasting results. Functional Magnetic Resonance Imaging (fMRI) showed that blue-wavelength (473 nm) light enhanced the responses to emotional stimuli in the amygdala as well as its crosstalk with the hypothalamus.⁷ In contrast, in a resting-state fMRI paradigm, exposure to a warm polychromatic white light (~2800K; 100 lux) suppressed amygdala activity compared to darkness (<1 lux) while its connectivity with the ventromedial prefrontal cortex was enhanced.¹⁵ Furthermore, a dose-dependent effect of three weeks of bright-light therapy was reported to reduce threat-related reactivity of the amygdala and to increase its functional connectivity with medial prefrontal cortex, suggesting these changes may be part of the mechanism that mediates the beneficial effects of bright-light therapy.¹⁶ The difference between studies may arise from differences in the light sources, from being engaged in a cognitive task vs. at rest and from considering shorter or longer impacts of light. They may also be due to the context-dependent response of the different nuclei of the amygdala that could not be addressed given the data resolution. Moving away from light’s impact, seasonal differences in the volume of subregions of the human amygdala were reported, with a peak in the summer. However, there was no association between mood measures and amygdala subregions volumes and photoperiod.¹⁷ “***

Results:

5. Although full methods are presented later, it would be helpful to specify the time frame of ‘morning’ here.

We now specify the following:

'The brain activity of 29 healthy young participants (24y ±3.1; 18 women) was recorded during an fMRI protocol conducted in Liège (Belgium) in the morning (scans 3 to 3.5hrs after habitual wake-up time),...' (line 119-120);

'Participants arrived at the laboratory 1.5h after wake-up time' (line 129 – figure 1 caption & line 513 - methods);

*'In total, 29 participants (24y ±3.1; 18 women) performed an executive (always first), an emotional and an attentional task (pseudo-randomly 2nd or 3rd, purple arrow; **meaning that the fMRI recording of the emotional task was completed 3 to 3.5h after wake-up time**).'* (line 133-134, caption figure 1);

'All participants included in this analysis were therefore scanned ~3h after habitual wake-up time and between 8 am and 1:15 pm [Fig.1A].' (line 521-523, methods).

6. Minor typo in the caption of Figure 1 – 'lose' sleep-wake schedule (loose?). The prior light history standardisation is also not clear, the text sounds more like it is light exposure across those 7 days, when in fact it is only the immediate light history prior to the scan.

We corrected the typo (line 136) and modified the figure caption as follows (line xx) *'Participants followed 7-days of regular **loose** sleep-wake schedule (+- 1h, verified using actigraphy) before the MRI scan. The prior light history of participants was standardised **on the morning of the MRI scan**, consisting of 5-min 1000 lux light and the following 45-min under dim (<10 lux) light.'* Line128-130.

7. It would be helpful to indicate local season on the graphs which show data by day of year (and/or actual changes in photoperiod).

We added coloured panels to the figure to represent the seasons. See Figure 2.H-I.

8. The way in which affective state was measured should be noted here.

We have clarified this aspect as follows (line xx): *We further assessed variation in the mood of our participants, **based on scores on the Beck Depression Inventory-II**,¹⁸ and...".* (line 163).

9. Many of the associations shown in the bottom section of Figure 2 were very weak. Given that some did not survive correction, the authors should show/state this more clearly on the figure.

We stress that, as stated in the text, the associations between time-of-year and the BMN and AAA as well as between mood and the medial and cortical nuclei and ATA survived

correction for multiple comparisons. We do not consider them weak even though effects may be small. As also stated in the text (line 413): “*The sizes of the effects we detected are small such that light exposure may be beneficial to the emotional state through a repeated effect on the BMN and medial parts of the amygdala (or on the prefrontal cortex).*”

We now make clearer in the figure captions that some associations did not survive correction for multiple testing. Line 238 – caption figure 2 ‘*A similar statistical trend (that did not survive correction for multiple comparisons) was detected in the anterior amygdaloid area (L; subpart 9) and the intercalated nuclei (M; subpart 10).*’ Lines 285 & 289 – caption of Figure 3: ‘*as indicated by a statistical trend (that did not survive correction for multiple comparisons).*’

Discussion:

10. “Affective state varies with seasons in a large portion of the healthy population” – given depression scores (the measure of ‘affective state’ used in this study) did not vary by season in this sample, there may be a better way to introduce the summary of your findings.

We have changed this sentence to the following “**Cognitive brain function varies with seasons, and psychiatric symptoms in patients also show seasonal variations.**^{19,20}” (Line 308-309).

11. The discussion overall is well-reasoned and written. However, at times, the results, particularly those pertaining to affective state and season, are overstated given the relatively weak relationships presented.

Thank you for your comment. We have edited the discussion so the results are presented in a more balanced way. For instance, we now state

(line 377-384): “**Light may impact mood through both the direct and indirect routes. Testing whether the timescale (acute vs long-term photoperiod) changes the pathways involved could be important for determining how light affects mood. The acute impact could be mainly mediated by a direct projection to the medial amygdala. In contrast, since seasonal affective disorder was found to be associated with a misalignment of the circadian clock that can be corrected by light therapy^{19,21}, the longer-term impact of photoperiod could indirectly reach the BMN through the SCN and/or the lateral hypothalamus. Future studies including connectivity analyses should test these hypotheses.**”

(line 385-392): “**Importantly, the impact of light on the amygdala was not uniform across its subparts at the different times of the year and with respect to mood (as indicated by**

significant interaction terms), at least when light is delivered in the morning. **Statistical trends potentially pointed towards the nuclei driving these non-uniform responses to light. The variation with time-of-year may be driven by the ATA, which showed a larger impact of light around summer solstice, while the variation related to worse mood may arise from the AAA. Future research should test whether these non-uniform responses to light contribute to the effectiveness of light therapy.**

Methods:

12. Minor typo/clarification needed re: “Subject willing to exit the scanner” – perhaps supposed to be wishing to exit?

We have changed this to ‘*subject **wanted to exit the scanner before the end of the session***’ - (line 502)

13. Additional information regarding the standardisation of prior light would be helpful – i.e., are these measures the maximal/minimal level in the room, how was 1000 lux delivered, using a light therapy device? Additional descriptions of the geometric space would be helpful to understand the actual retinal light exposure. Additionally, the specification of the lighting in the MRI room should be included.

We have added the following into the methods section (line 515-518): “To standardise participants' recent light history, they were exposed to 5min of bright white light (1000 lux; **with the chin on a chin-rest, ~15cm away from a plastic diffuser in front of a polychromatic halogen light bulb**) and were then maintained in dim light (<10 lux) for 45min (**bright and dim light levels were controlled for each participants at eye level**).”

14. Can the authors speak to their decision to use the BDI as a marker of ‘affective state’? They also had BAI available, which is arguably also relevant to the regions of interest, and appears to show a greater range of scores, but another measure which may be more sensitive to normal mood variation in healthy persons would have likely been more appropriate than either. I am concerned that the limited range of BDI scores may not meaningfully reflect differences in ‘affective state’ in this population. It would be more appropriate to refer to this measure as depressive symptoms, rather than affective state, throughout the paper. At a minimum the actual range of scores and mean/SD should be reported here in the methods, but ideally the limitations of using this measure in an otherwise healthy population should also be discussed in the paper.

We have edited the paper so ‘affective state’ has been edited to ‘mood’ and we clearly mention at first that it is assessed through the BDI. This term is preferred to depressive symptoms as the use of this terminology could lead the reader to think that we included patients when all our participants were healthy and free of psychiatric diagnosis. The reviewer is right when stating that BAI could also be a measure of interest. BAI has however been reported to be strongly associated with BDI and as its name indicate is related to anxiety rather than mood which is truly what we are interested in given that light therapy and photoperiod are most known to affect mood disorders and subclinical mood variations. **“Furthermore, we used BDI to measure mood in a young healthy population, implying non-clinical variations in mood scores. We cannot generalise our results to populations with depression or other psychiatric conditions.”** (line 464-466)

References

1. Sander, D. *et al.* Emotion and attention interactions in social cognition: Brain regions involved in processing anger prosody. *Neuroimage* **28**, 848–858 (2005).
2. Grandjean, D. *et al.* The voices of wrath: Brain responses to angry prosody in meaningless speech. *Nat Neurosci* **8**, 145–146 (2005).
3. Campbell, I. *et al.* Regional response to light illuminance across the human hypothalamus. *Elife* **13**, (2024).
4. Campbell, I. *et al.* Impact of light on task-evoked pupil responses during cognitive tasks. *J Sleep Res* **33**, (2024).
5. Beckers, E. *et al.* Impact of repeated short light exposures on sustained pupil responses in an fMRI environment. *J Sleep Res* **33**, (2024).
6. Paparella, I. *et al.* Light modulates task-dependent thalamo-cortical connectivity during an auditory attentional task. *Commun Biol* **6**, (2023).
7. Vandewalle, G. *et al.* Spectral quality of light modulates emotional brainresponses in humans. *Proc Natl Acad Sci U S A* **107**, 19549–19554 (2010).
8. Vandewalle, G. *et al.* Abnormal Hypothalamic Response to Light in Seasonal Affective Disorder. *Biol Psychiatry* **70**, 954–961 (2011).
9. Allen, A. E. *et al.* Melanopsin-driven light adaptation in mouse vision. *Current Biology* **24**, 2481–2490 (2014).
10. Makaronidis, J. M. & Batterham, R. L. Obesity, body weight regulation and the brain: Insights from fMRI. *British Journal of Radiology* **91**, (2018).

11. Whalen, P. J. & Phelps, E. A. *The Human Amygdala. The Human Amygdala* (Guilford Press, 2009).
12. Fisher, P. M. *et al.* Three-week bright-light intervention has dose-related effects on threat-related corticolimbic reactivity and functional coupling. *Biol Psychiatry* **76**, 332–339 (2014).
13. Huang, L. *et al.* A Visual Circuit Related to Habenula Underlies the Antidepressive Effects of Light Therapy. *Neuron* **102**, 128-142.e8 (2019).
14. An, K. *et al.* A circadian rhythm-gated subcortical pathway for nighttime-light-induced depressive-like behaviors in mice. *Nat Neurosci* **23**, 869–880 (2020).
15. McGlashan, E. M., Poudel, G. R., Jamadar, S. D., Phillips, A. J. K. & Cain, S. W. Afraid of the dark: Light acutely suppresses activity in the human amygdala. *PLoS One* **16**, (2021).
16. Fisher, P. M. *et al.* Three-week bright-light intervention has dose-related effects on threat-related corticolimbic reactivity and functional coupling. *Biol Psychiatry* **76**, 332–339 (2014).
17. Majrashi, N. A., Alyami, A. S., Shubayr, N. A., Alenezi, M. M. & Waiter, G. D. Amygdala and subregion volumes are associated with photoperiod and seasonal depressive symptoms: A cross-sectional study in the UK Biobank cohort. *European Journal of Neuroscience* **55**, 1388–1404 (2022).
18. Beck, A. T., Ward, C. H., Mendelson, M., Mock, J. & Erbaugh, J. An inventory for measuring depression. *Arch Gen Psychiatry* **4**, 561–571 (1961).
19. Zhang, R. & Volkow, N. D. Seasonality of brain function: role in psychiatric disorders. *Translational Psychiatry* 2023 13:1 **13**, 1–11 (2023).
20. Meyer, C. *et al.* Seasonality in human cognitive brain responses. *Proc Natl Acad Sci U S A* **113**, 3066–3071 (2016).
21. Lavoie, M. P. *et al.* Evidence of a Biological Effect of Light Therapy on the Retina of Patients with Seasonal Affective Disorder. *Biol Psychiatry* **66**, 253–258 (2009).

Reviewer #1 (Remarks to the Author):

The authors have addressed my concerns, and the revised manuscript is significantly improved. I have no further comments.

Response : We would like to thank the reviewer #1 for taking the time to review our manuscript.

Reviewer #2 (Remarks to the Author):

The authors have thoughtfully revised their manuscript. One minor note regarding the new figure panel (Fig 3f) - the embedded figure does not appear to be the new version. I would also suggest adding the local season "names" to the figure in addition to colours for ease of interpretation.

Response : We would like to thank the reviewer #2 for taking the time to review our manuscript. We have edited the combined Figure 3 so it includes the updated version of Fig.3F. We have included the following - '*Seasons represented by background colours (winter – blue, spring – green, summer – yellow, autumn – orange)*' - in the figure legend for Figure 2H,I and 3F, so it is easier to interpret.